# NEURAL PERSISTENCE: A COMPLEXITY MEASURE FOR DEEP NEURAL NETWORKS USING ALGEBRAIC TOPOLOGY

**Bastian Rieck**[1,2,†]**, Matteo Togninalli**[1,2,†]**, Christian Bock**[1,2,†]**,**
**Michael Moor**[1,2]**, Max Horn**[1,2]**, Thomas Gumbsch**[1,2]**, Karsten Borwardt**[1,2]
[1]DEPARTMENT OF BIOSYSTEMS SCIENCE AND ENGINEERING, ETH ZURICH, SWITZERLAND
[2]SIB SWISS INSTITUTE OF BIOINFORMATICS, SWITZERLAND
[†]These authors contributed equally

## ABSTRACT

While many approaches to make neural networks more fathomable have been proposed, they are restricted to interrogating the network with input data. Measures for characterizing and monitoring structural properties, however, have not been developed. In this work, we propose *neural persistence*, a complexity measure for neural network architectures based on topological data analysis on weighted stratified graphs. To demonstrate the usefulness of our approach, we show that *neural persistence* reflects best practices developed in the deep learning community such as dropout and batch normalization. Moreover, we derive a neural persistence-based stopping criterion that shortens the training process while achieving comparable accuracies as early stopping based on validation loss.

## 1 INTRODUCTION

The practical successes of deep learning in various fields such as image processing (Simonyan & Zisserman, 2015; He et al., 2016; Hu et al., 2018), biomedicine (Ching et al., 2018; Rajpurkar et al., 2017; Rajkomar et al., 2018), and language translation (Bahdanau et al., 2015; Sutskever et al., 2014; Wu et al., 2016) still outpace our theoretical understanding. While hyperparameter adjustment strategies exist (Bengio, 2012), formal measures for assessing the generalization capabilities of deep neural networks have yet to be identified (Zhang et al., 2017). Previous approaches for improving theoretical and practical comprehension focus on interrogating networks with input data. These methods include i) feature visualization of deep convolutional neural networks (Zeiler & Fergus, 2014; Springenberg et al., 2015), ii) sensitivity and relevance analysis of features (Montavon et al., 2017), iii) a descriptive analysis of the training process based on information theory (Tishby & Zaslavsky, 2015; Shwartz-Ziv & Tishby, 2017; Saxe et al., 2018; Achille & Soatto, 2018), and iv) a statistical analysis of interactions of the learned weights (Tsang et al., 2018). Additionally, Raghu et al. (2017) develop a measure of *expressivity* of a neural network and use it to explore the empirical success of batch normalization, as well as for the definition of a new regularization method. They note that one key challenge remains, namely to provide meaningful insights while maintaining theoretical generality. This paper presents a method for elucidating neural networks in light of both aspects.

We develop *neural persistence*, a novel measure for characterizing neural network structural complexity. In doing so, we adopt a new perspective that integrates both network weights and connectivity while not relying on interrogating networks through input data. Neural persistence builds on computational techniques from algebraic topology, specifically topological data analysis (TDA), which was already shown to be beneficial for feature extraction in deep learning (Hofer et al., 2017) and describing the complexity of GAN sample spaces (Khrulkov & Oseledets, 2018). More precisely, we rephrase deep networks with fully-connected layers into the language of algebraic topology and develop a measure for assessing the structural complexity of i) individual layers, and ii) the entire network. In this work, we present the following contributions:

- We introduce *neural persistence*, a novel measure for characterizing the structural complexity of neural networks that can be efficiently computed.
- We prove its theoretical properties, such as upper and lower bounds, thereby arriving at a normalization for comparing neural networks of varying sizes.
- We demonstrate the practical utility of neural persistence in two scenarios: i) it correctly captures the benefits of dropout and batch normalization during the training process, and ii) it can be easily used as a competitive early stopping criterion that does not require validation data.

## 2 BACKGROUND: TOPOLOGICAL DATA ANALYSIS

Topological data analysis (TDA) recently emerged as a field that provides computational tools for analysing complex data within a rigorous mathematical framework that is based on *algebraic topology*. This paper uses persistent homology, a theory that was developed to understand high-dimensional manifolds (Edelsbrunner et al., 2002; Edelsbrunner & Harer, 2010), and has since been successfully employed in characterizing graphs (Sizemore et al., 2017; Rieck et al., 2018), finding relevant features in unstructured data (Lum et al., 2013), and analysing image manifolds (Carlsson et al., 2008). This section gives a brief summary of the key concepts; please refer to Edelsbrunner & Harer (2010) for an extensive introduction.

**Simplicial homology**   The central object in algebraic topology is a simplicial complex K, i.e. a high-dimensional generalization of a graph, which is typically used to describe complex objects such as manifolds. Various notions to describe the connectivity of K exist, one of them being simplicial homology. Briefly put, simplicial homology uses matrix reduction algorithms (Munkres, 1996) to derive a set of groups, the homology groups, for a given simplicial complex K. Homology groups describe topological features—colloquially also referred to as holes—of a certain dimension $d$, such as connected components ($d = 0$), tunnels ($d = 1$), and voids ($d = 2$). The information from the $d$th homology group is summarized in a simple complexity measure, the $d$th Betti number $\beta_d$, which merely counts the number of $d$-dimensional features: a circle, for example, has Betti numbers $(1, 1)$, i.e. one connected component and one tunnel, while a filled circle has Betti numbers $(1, 0)$, i.e. one connected component but no tunnel. In the context of analysing simple feedforward neural networks for two classes, Bianchini & Scarselli (2014) calculated bounds of Betti numbers of the decision region belonging to the positive class, and were thus able to show the implications of different activation functions. These ideas were extended by Guss & Salakhutdinov (2018) to obtain a measure of the topological complexity of decision boundaries.

**Persistent homology**   For the analysis of real-world data sets, however, Betti numbers turn out to be of limited use because their representation is too coarse and unstable. This prompted the development of persistent homology. Given a simplicial complex K with an additional set of weights $a_0 \leq a_1 \leq \cdots \leq a_{m-1} \leq a_m$, which are commonly thought to represent the idea of a scale, it is possible to put K in a filtration, i.e. a nested sequence of simplicial complexes $\emptyset = K_0 \subseteq K_1 \subseteq \cdots \subseteq K_{m-1} \subseteq K_m = K$. This filtration is thought to represent the 'growth' of K as the scale is being changed. During this growth process, topological features can be *created* (new vertices may be added, for example, which creates a new connected component) or *destroyed* (two connected components may merge into one). Persistent homology tracks these changes and represents the creation and destruction of a feature as a point $(a_i, a_j) \in \mathbb{R}^2$ for indices $i \leq j$ with respect to the filtration. The collection of all points corresponding to $d$-dimensional topological features is called the $d$th persistence diagram $\mathcal{D}_d$. It can be seen as a collection of Betti numbers at multiple scales. Given a point $(x, y) \in \mathcal{D}_d$, the quantity $\mathrm{pers}(x, y) := |y - x|$ is referred to as its *persistence*. Typically, high persistence is considered to correspond to features, while low persistence is considered to indicate noise (Edelsbrunner et al., 2002).

## 3 A NOVEL MEASURE FOR NEURAL NETWORK COMPLEXITY

This section details *neural persistence*, our novel measure for assessing the structural complexity of neural networks. By exploiting both network structure and weight information through persistent homology, our measure captures network expressiveness and goes beyond mere connectivity properties. Subsequently, we describe its calculation, provide theorems for theoretical and empirical

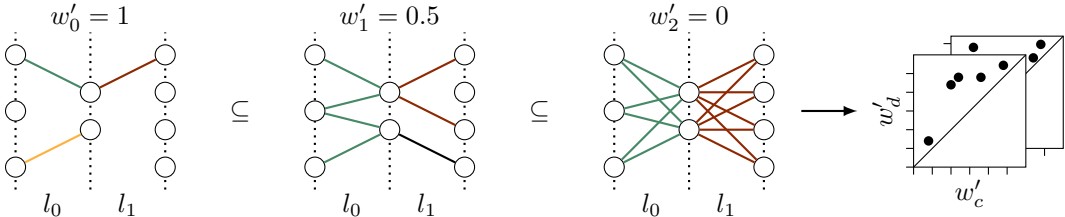

Figure 1: Illustrating the neural persistence calculation of a network with two layers ($l_0$ and $l_1$). Colours indicate connected components per layer. The filtration process is depicted by colouring connected components that are created or merged when the respective weights are greater than or equal to the threshold $w'_i$. As $w'_i$ decreases, network connectivity increases. Creation and destruction thresholds are collected in one persistence diagram per layer (right), and summarized according to Equation 1 for calculating neural persistence.

bounds, and show the existence of neural networks complexity regimes. To summarize this section, Figure 1 illustrates how our method treats a neural network.

## 3.1 NEURAL PERSISTENCE

Given a feedforward neural network with an arrangement of neurons and their connections $E$, let $\mathcal{W}$ refer to the set of weights. Since $\mathcal{W}$ is typically changing during training, we require a function $\varphi \colon E \to \mathcal{W}$ that maps a specific edge to a weight. Fixing an activation function, the connections form a *stratified graph*.

**Definition 1** (Stratified graph and layers). *A stratified graph is a multipartite graph $G = (V, E)$ satisfying $V = V_0 \sqcup V_1 \sqcup \ldots$, such that if $u \in V_i$, $v \in V_j$, and $(u, v) \in E$, we have $j = i + 1$. Hence, edges are only permitted between adjacent vertex sets. Given $k \in \mathbb{N}$, the kth layer of a stratified graph is the unique subgraph $G_k := (V_k \sqcup V_{k+1}, E_k := E \cap \{V_k \times V_{k+1}\})$.*

This enables calculating the persistent homology of $G$ and each $G_k$, using the filtration induced by sorting all weights, which is common practice in topology-based network analysis (Carstens & Horadam, 2013; Horak et al., 2009) where weights often represent closeness or node similarity. However, our context requires a novel filtration because the weights arise from an incremental fitting procedure, namely the training, which could theoretically lead to unbounded values. When analysing geometrical data with persistent homology, one typically selects a filtration based on the (Euclidean) distance between data points (Bubenik, 2015). The filtration then connects points that are increasingly distant from each other, starting from points that are direct neighbours. Our network filtration aims to mimic this behaviour in the context of fully-connected neural networks. Our framework does not *explicitly* take activation functions into account; however, activation functions influence the evolution of weights during training.

**Filtration** Given the set of weights $\mathcal{W}$ for one training step, let $w_{\max} := \max_{w \in \mathcal{W}} |w|$. Furthermore, let $\mathcal{W}' := \{|w|/w_{\max} \mid w \in \mathcal{W}\}$ be the set of transformed weights, indexed in non-ascending order, such that $1 = w'_0 \geq w'_1 \geq \cdots \geq 0$. This permits us to define a filtration for the kth layer $G_k$ as $G_k^{(0)} \subseteq G_k^{(1)} \subseteq \ldots$, where $G_k^{(i)} := (V_k \sqcup V_{k+1}, \{(u, v) \mid (u, v) \in E_k \wedge \varphi'(u, v) \geq w'_i\})$ and $\varphi'(u, v) \in \mathcal{W}'$ denotes the transformed weight of an edge. We tailored this filtration towards the analysis of neural networks, for which large (absolute) weights indicate that certain neurons exert a larger influence over the final activation of a layer. The strength of a connection is thus preserved by the filtration, and weaker weights with $|w| \approx 0$ remain close to 0. Moreover, since $w' \in [0, 1]$ holds for the transformed weights, this filtration makes the network invariant to scaling, which simplifies the comparison of different networks.

**Persistence diagrams** Having set up the filtration, we can calculate persistent homology for every layer $G_k$. As the filtration contains at most 1-simplices (edges), we capture zero-dimensional topological information, i.e. how connected components are created and merged during the filtration. These information are structurally equivalent to calculating a maximum spanning tree using the

---

**Algorithm 1** Neural persistence calculation

---

**Require:** Neural network with $l$ layers and weights $\mathcal{W}$

1: $w_{\max} \leftarrow \max_{w \in \mathcal{W}} |w|$        ▷ Determine largest absolute weight
2: $\mathcal{W}' \leftarrow \{|w|/w_{\max} \mid w \in \mathcal{W}\}$        ▷ Transform weights for filtration
3: **for** $k \in \{0, \dots, l-1\}$ **do**
4:     $F_k \leftarrow G_k^{(0)} \subseteq G_k^{(1)} \subseteq \dots$        ▷ Establish filtration of $k$th layer
5:     $\mathcal{D}_k \leftarrow \text{PERSISTENTHOMOLOGY}(F_k)$        ▷ Calculate persistence diagram
6: **end for**
7: **return** $\{\|\mathcal{D}_0\|_p, \dots, \|\mathcal{D}_{l-1}\|_p\}$        ▷ Calculate neural persistence for each layer

---

weights, or performing hierarchical clustering with a specific setup (Carlsson & Mémoli, 2010). While it would theoretically be possible to include higher-dimensional information about each layer $G_k$, for example in the form of cliques (Rieck et al., 2018), we focus on zero-dimensional information in this paper, because of the following advantages: i) the resulting values are easily interpretable as they essentially describe the clustering of the network at multiple weight thresholds, ii) previous research (Rieck & Leitte, 2016; Hofer et al., 2017) indicates that zero-dimensional topological information is already capturing a large amount of information, and iii) persistent homology calculations are highly efficient in this regime (see below). We thus calculate zero-dimensional persistent homology with this filtration. The resulting persistence diagrams have a special structure: since our filtration solely sorts *edges*, all vertices are present at the beginning of the filtration, i.e. they are already part of $G_k^{(0)}$ for each $k$. As a consequence, they are assigned a weight of 1, resulting in $|V_k \times V_{k+1}|$ connected components. Hence, entries in the corresponding persistence diagram $\mathcal{D}_k$ are of the form $(1, x)$, with $x \in \mathcal{W}'$, and will be situated *below* the diagonal, similar to superlevel set filtrations (Bubenik, 2015; Cohen-Steiner et al., 2009). Using the $p$-norm of a persistence diagram, as introduced by Cohen-Steiner et al. (2010), we obtain the following definition for neural persistence.

**Definition 2** (Neural persistence). *The neural persistence of the $k$th layer $G_k$, denoted by $\text{NP}(G_k)$, is the $p$-norm of the persistence diagram $\mathcal{D}_k$ resulting from our previously-introduced filtration, i.e.*

$$\text{NP}(G_k) := \|\mathcal{D}_k\|_p := \Big( \sum_{(c,d) \in \mathcal{D}_k} \text{pers}(c,d)^p \Big)^{\frac{1}{p}}, \tag{1}$$

*which (for $p = 2$) captures the Euclidean distance of points in $\mathcal{D}_k$ to the diagonal.*

The $p$-norm is known to be a stable summary (Cohen-Steiner et al., 2010) of topological features in a persistence diagram. For neural persistence to be a meaningful measure of structural complexity, it should increase as a neural network is learning. We evaluate this and other properties in Section 4.

Algorithm 1 provides pseudocode for the calculation process. It is highly efficient: the filtration (line 4) amounts to sorting all $n$ weights of a network, which has a computational complexity of $\mathcal{O}(n \log n)$. Calculating persistent homology of this filtration (line 5) can be realized using an algorithm based on union–find data structures Edelsbrunner et al. (2002). This has a computational complexity of $\mathcal{O}(n \cdot \alpha(n))$, where $\alpha(\cdot)$ refers to the extremely slow-growing inverse of the Ackermann function (Cormen et al., 2009, Chapter 22). We make our implementation and experiments available under `https://github.com/BorgwardtLab/Neural-Persistence`.

## 3.2 PROPERTIES OF NEURAL PERSISTENCE

We elucidate properties about neural persistence to permit the comparison of networks with different architectures. As a first step, we derive *bounds* for the neural persistence of a single layer $G_k$.

**Theorem 1.** *Let $G_k$ be a layer of a neural network according to Definition 1. Furthermore, let $\varphi_k \colon E_k \to \mathcal{W}'$ denote the function that assigns each edge of $G_k$ a transformed weight. Using the filtration from Section 3.1 to calculate persistent homology, the neural persistence $\text{NP}(G_k)$ of the $k$th layer satisfies*

$$0 \le \text{NP}(G_k) \le \Big( \max_{e \in E_k} \varphi_k(e) - \min_{e \in E_k} \varphi_k(e) \Big) (|V_k \times V_{k+1}| - 1)^{\frac{1}{p}}, \tag{2}$$

*where $|V_k \times V_{k+1}|$ denotes the cardinality of the vertex set, i.e. the number of neurons in the layer.*

*Proof.* We prove this constructively and show that the bounds can be realized. For the lower bound, let $G_k^-$ be a fully-connected layer with $|V_k|$ vertices and, given $\theta \in [0, 1]$, let $\varphi_k(e) := \theta$ for every edge $e$. Since a vertex $v$ is created before its incident edges, the filtration degenerates to a lexicographical ordering of vertices and edges, and all points in $\mathcal{D}_k$ will be of the form $(\theta, \theta)$. Thus, $\text{NP}(G_k^-) = 0$. For the upper bound, let $G_k^+$ again be a fully-connected layer with $|V_k| \geq 3$ vertices and let $a, b \in [0, 1]$ with $a < b$. Select one edge $e'$ at random and define a weight function as $\varphi(e') := b$ and $\varphi(e) := a$ otherwise. In the filtration, the addition of the first edge will create a pair of the form $(b, b)$, while all other pairs will be of the form $(b, a)$. Consequently, we have

$$\text{NP}(G_k^+) = \left( \text{pers}(b, b)^p + (n-1) \cdot \text{pers}(b, a)^p \right)^{\frac{1}{p}} = (b - a) \cdot (n-1)^{\frac{1}{p}} \tag{3}$$

$$= \left( \max_{e \in E_k} \varphi(e) - \min_{e \in E_k} \varphi(e) \right) (|V_k| - 1)^{\frac{1}{p}}, \tag{4}$$

so our upper bound can be realized. To show that this term cannot be exceeded by $\text{NP}(G)$ for any $G$, suppose we perturb the weight function $\widetilde{\varphi}(e) := \varphi(e) + \epsilon \in [0, 1]$. This cannot increase NP, however, because each difference $b - a$ in Equation 3 is maximized by $\max \varphi(e) - \min \varphi(e)$. $\quad\square$

We can use the upper bound of Theorem 1 to normalize the neural persistence of a layer, making it possible to compare layers (and neural networks) that feature different architectures, i.e. a different number of neurons.

**Definition 3** (Normalized neural persistence). *For a layer $G_k$ following Definition 1, using the upper bound of Theorem 1, the normalized neural persistence $\widetilde{\text{NP}}(G_k)$ is defined as the neural persistence of $G_k$ divided by its upper bound, i.e. $\widetilde{\text{NP}}(G_k) := \text{NP}(G_k) \cdot \text{NP}(G_k^+)^{-1}$.*

The normalized neural persistence of a layer permits us to extend the definition to an entire network. While this is more complex than using a single filtration for a neural network, this permits us to side-step the problem of different layers having different scales.

**Definition 4** (Mean normalized neural persistence). *Considering a network as a stratified graph $G$ according to Definition 1, we sum the neural persistence values per layer to obtain the* mean normalized neural persistence, *i.e. $\overline{\text{NP}}(G) := 1/l \cdot \sum_{k=0}^{l-1} \widetilde{\text{NP}}(G_k)$.*

While Theorem 1 gives a lower and upper bound in a general setting, it is possible to obtain empirical bounds when we consider the tuples that result from the computation of a persistence diagram. Recall that our filtration ensures that the persistence diagram of a layer contains tuples of the form $(1, w_i)$, with $w_i \in [0, 1]$ being a transformed weight. Exploiting this structure permits us to obtain bounds that could be used prior to calculating the actual neural persistence value in order to make the implementation more efficient.

**Theorem 2.** *Let $G_k$ be a layer of a neural network as in Theorem 1 with $n$ vertices and $m$ edges whose edge weights are sorted in non-descending order, i.e. $w_0 \leq w_2 \leq \cdots \leq w_{m-1}$. Then $\text{NP}(G_k)$ can be empirically bounded by*

$$\left\| \mathbb{1} - \mathbf{w}_{\max} \right\|_p \leq \text{NP}(G_k) \leq \left\| \mathbb{1} - \mathbf{w}_{\min} \right\|_p, \tag{5}$$

*where $\mathbf{w}_{\max} = (w_{m-1}, w_{m-2}, \ldots, w_{m-n})^T$ and $\mathbf{w}_{\min} = (w_0, w_2, \ldots, w_{n-1})^T$ are the vectors containing the $n$ largest and $n$ smallest weights, respectively.*

*Proof.* See Section A.2 in the appendix.

**Complexity regimes in neural persistence**   As an application of the two theorems, we briefly take a look at how neural persistence changes for different classes of simple neural networks. To this end, we train a perceptron on the 'MNIST' data set. Since our measure uses the weight matrix of a perceptron, we can compare its neural persistence with the neural persistence of random weight matrices, drawn from different distributions. Moreover, we can compare trained networks with respect to their initial parameters. Figure 2 depicts the neural persistence values as well as the lower bounds according to Theorem 2 for different settings. We can see that a network in which the optimizer diverges (due to improperly selected parameters) is similar to a random Gaussian matrix.

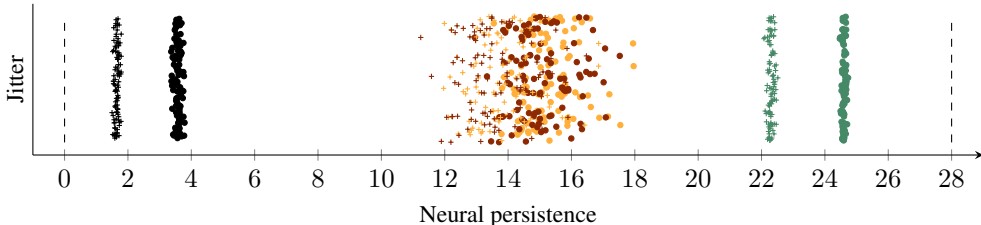

Figure 2: Neural persistence values of trained perceptrons (green), diverging ones (yellow), random Gaussian matrices (red), and random uniform matrices (black). We performed 100 runs per category; dots indicate neural persistence while crosses indicate the predicted lower bound according to Theorem 2. The bounds according to Theorem 1 are shown as dashed lines.

Trained networks, on the other hand, are clearly distinguished from all other networks. Uniform matrices have a significantly lower neural persistence than Gaussian ones. This is in line with the intuition that the latter type of networks induces functional sparsity because few neurons have large absolute weights. For clarity, we refrain from showing the empirical upper bounds because most weight distributions are highly right-tailed; the bound will not be as tight as the lower bound. These results are in line with a previous analysis (Sizemore et al., 2017) of small weighted networks, in which persistent homology is seen to outperform traditional graph-theoretical complexity measures such as the clustering coefficient (see also Section A.1 in the appendix). For deeper networks, additional experiments discuss the relation between validation accuracy and neural persistence (Section A.5), the impact of different data distributions, as well as the variability of neural persistence for architectures of varying depth (Section A.6).

## 4 EXPERIMENTS

This section demonstrates the utility and relevance of neural persistence for fully connected deep neural networks. We examine how commonly used regularization techniques (batch normalization and dropout) affect neural persistence of trained networks. Furthermore, we develop an early stopping criterion based on neural persistence and we compare it to the traditional criterion based on validation loss. We used different architectures with *ReLU* activation functions across experiments. The brackets denote the number of units per *hidden* layer. In addition, the Adam optimizer with hyperparameters tuned via cross-validation was used unless noted otherwise. Please refer to Table A.1 in the appendix for further details about the experiments.

### 4.1 DEEP LEARNING BEST PRACTICES IN LIGHT OF NEURAL PERSISTENCE

We compare the mean normalized neural persistence (see Definition 4) of a two-layer (with an architecture of $[650, 650]$) neural network to two models where batch normalization (Ioffe & Szegedy, 2015) or dropout (Srivastava et al., 2014) are applied. Figure 3 shows that the networks designed according to best practices yield higher normalized neural persistence values on the 'MNIST' data set in comparison to an unmodified network. The effect of dropout on the mean normalized neural persistence is more pronounced and this trend is directly analogous to the observed accuracy on

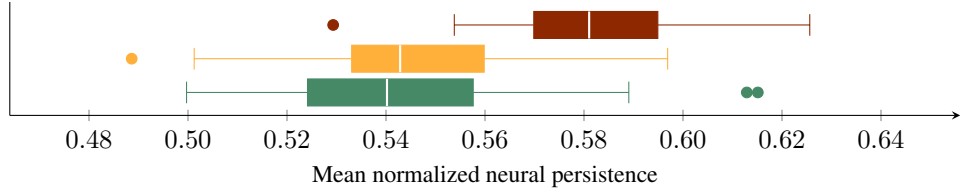

Figure 3: Comparison of mean normalized neural persistence for trained networks without modifications (green), with batch normalization (yellow), and with 50% of the neurons dropped out during training (red) for the 'MNIST' data set (50 runs per setting).

the test set. These results are consistent with expectations if we consider dropout to be similar to ensemble learning (Hara et al., 2016). As individual parts of the network are trained independently, a higher degree of per-layer redundancy is expected, resulting in a different structural complexity. Overall, these results indicate that for a fixed architecture approaches targeted at increasing the neural persistence during the training process may be of particular interest.

## 4.2 EARLY STOPPING BASED ON NEURAL PERSISTENCE

Neural persistence can be used as an *early stopping* criterion that does not require a validation data set to prevent overfitting: if the mean normalized neural persistence does not increase by more than $\Delta_{\min}$ during a certain number of epochs $g$, the training process is stopped. This procedure is called 'patience' and Algorithm 2 describes it in detail. A similar variant of this algorithm, using validation loss instead of persistence, is the state-of-the-art for early stopping in training (Bengio, 2012; Chollet et al., 2015). To evaluate the efficacy of our measure, we compare it against validation loss in an extensive set of scenarios. More precisely, for a training process with at most $G$ epochs, we define a $G \times G$ parameter grid consisting of the 'patience' parameter $g$ and a burn-in rate $b$ (both measured in epochs). $b$ defines the number of epochs after which an early stopping criterion starts monitoring, thereby preventing underfitting. Subsequently, we set $\Delta_{\min} = 0$ for all measures to remain comparable and scale-invariant, as non-zero values could implicitly favour one of them due to scaling. For each data set, we perform 100 training runs of the same architecture, monitoring validation loss and mean normalized neural persistence every quarter epoch. The early stopping behaviour of both measures is simulated for each combination of $b$ and $g$ and their performance over all runs is summarized in terms of median test accuracy and median stopping epoch; if a criterion is not triggered for one run, we report the test accuracy at the end of the training and the number of training epochs. This results in a scatterplot, where each point (corresponding to a single parameter combination) shows the difference in epochs and the absolute difference in test accuracy (measured in percent). The quadrants permit an intuitive explanation: $Q_2$, for example, contains all configurations for which our measure stops *earlier*, while achieving a *higher* accuracy. Since $b$ and $g$ are typically chosen to be small in an early stopping scenario, we use grey points to indicate uncommon configurations for which $b$ or $g$ is larger than half of the total number of epochs. Furthermore, to summarize the performance of our measure, we calculate the barycentre of all configurations (green square).

Figure 4a depicts the comparison with validation loss for the 'Fashion-MNIST' (Xiao et al., 2017) data set; please refer to Section A.3 in the appendix for more data sets. Here, we observe that most common configurations are in $Q_2$ or in $Q_3$, i.e our criterion stops earlier. The barycentre is at $(-0.53, -0.08)$, showing that out of 625 configurations, on average we stop half an epoch earlier than validation loss, while losing virtually no accuracy (0.08%). Figure 4c depicts detailed differences in accuracy and epoch for our measure when compared to validation loss; each cell in a heatmap corresponds to a single parameter configuration of $b$ and $g$. In the heatmap of accuracy differences, blue, white, and red represent parameter combinations for which we obtain *higher, equal, or lower* accuracy, respectively, than with validation loss for the same parameters. Similarly, in the

---

**Algorithm 2** Early stopping based on mean normalized neural persistence

---

**Require:** Weighted neural network $\mathcal{N}$, patience $g$, $\Delta_{\min}$
 1: $P \leftarrow 0, G \leftarrow 0$            ▷ Initialize highest observed value and patience counter
 2: **procedure** EARLYSTOPPING($\mathcal{N}, g, \Delta_{\min}$)     ▷ Callback that monitors training at every epoch
 3:      $P' \leftarrow \overline{\text{NP}}(\mathcal{N})$
 4:      **if** $P' > P + \Delta_{\min}$ **then**     ▷ Update mean normalized neural persistence and reset counter
 5:          $P \leftarrow P', G \leftarrow 0$
 6:      **else**                                              ▷ Update patience counter
 7:          $G \leftarrow G + 1$
 8:      **end if**
 9:      **if** $G \geq g$ **then**                            ▷ Patience criterion has been triggered
10:          **return** $P$             ▷ Stop training and return highest observed value
11:      **end if**
12: **end procedure**

---

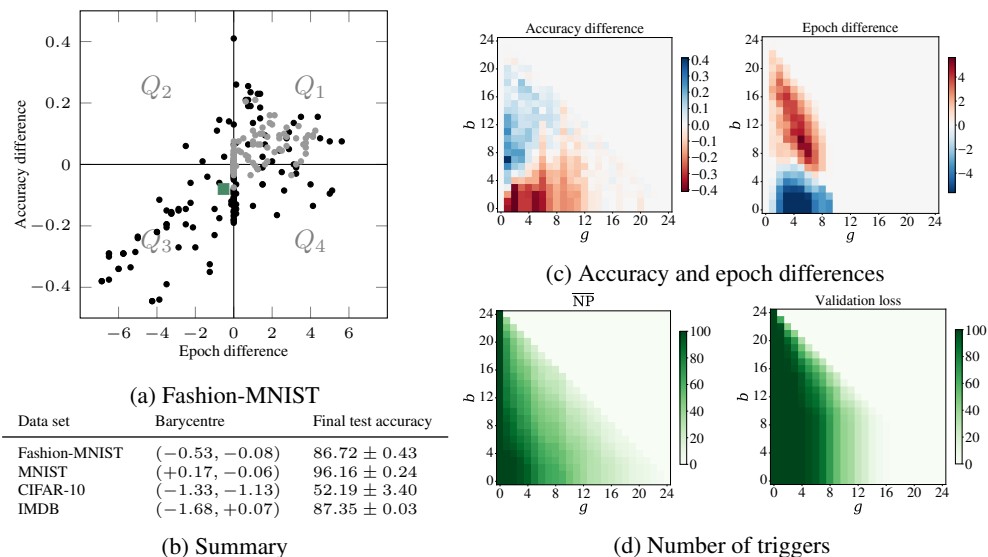

(a) Fashion-MNIST

| Data set | Barycentre | Final test accuracy |
|---|---|---|
| Fashion-MNIST | $(-0.53, -0.08)$ | $86.72 \pm 0.43$ |
| MNIST | $(+0.17, -0.06)$ | $96.16 \pm 0.24$ |
| CIFAR-10 | $(-1.33, -1.13)$ | $52.19 \pm 3.40$ |
| IMDB | $(-1.68, +0.07)$ | $87.35 \pm 0.03$ |

(b) Summary

(c) Accuracy and epoch differences

(d) Number of triggers

Figure 4: The visualizations depict the differences in accuracy and epoch for all comparison scenarios of mean normalized neural persistence versus validation loss, while the table summarizes the results on other data sets. Final test accuracies are shown irrespectively of early stopping to put the accuracy differences into context.

heatmap of epoch differences, green represents parameter combinations for which we stop *earlier* than validation loss. For $b \leq 8$, we stop earlier (0.62 epochs on average), while losing only $0.06\%$ accuracy. Finally, Figure 4d shows how often each measure is triggered. Ideally, each measure should consist of a dark green triangle, as this would indicate that *each* configuration stops all the time. For this data set, we observe that our method stops for more parameter combinations than validation loss, but not as frequently for all of them. To ensure comparability across scenarios, we did not use the validation data as additional training data when stopping with neural persistence; we refer to Section A.7 for additional experiments in data scarcity scenarios. We observe that our method stops earlier when overfitting can occur, and it stops later when longer training is beneficial.

## 5 DISCUSSION

In this work, we presented *neural persistence*, a novel topological measure of the structural complexity of deep neural networks. We showed that this measure captures topological information that pertains to deep learning performance. Being rooted in a rich body of research, our measure is theoretically well-defined and, in contrast to previous work, generally applicable as well as computationally efficient. We showed that our measure correctly identifies networks that employ best practices such as dropout and batch normalization. Moreover, we developed an early stopping criterion that exhibits competitive performance while not relying on a separate validation data set. Thus, by saving valuable data for training, we managed to boost accuracy, which can be crucial for enabling deep learning in regimes of smaller sample sizes. Following Theorem 2, we also experimented with using the $p$-norm of *all* weights of the neural network as a proxy for neural persistence. However, this did not yield an early stopping measure because it was never triggered, thereby suggesting that neural persistence captures salient information that would otherwise be hidden among all the weights of a network. We extended our framework to convolutional neural networks (see Section A.4) by deriving a closed-form approximation, and observed that an early stopping criterion based on neural persistence for convolutional layers will require additional work. Furthermore, we conjecture that assessing dissimilarities of networks by means of persistence diagrams (making use of higher-dimensional topological features), for example, will lead to further insights regarding their generalization and learning abilities. Another interesting avenue for future research would concern the analysis of the 'function space' learned by a neural network. On a more general level, *neural persistence* demonstrates the great potential of topological data analysis in machine learning.

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

## A  APPENDIX

### A.1  COMPARISON WITH GRAPH-THEORETICAL MEASURES

Traditional complexity/structural measures from graph theory, such as the clustering coefficient, the average shortest path length, and global/local efficiency are already known to be insufficiently accurate to characterize different models of complex random networks Sizemore et al. (2017). Our experiments indicate that this holds true for (deep) neural networks, too. As a brief example, we trained a perceptron on the MNIST data set with batch stochastic gradient descent ($\eta = 0.5$), achieving a test accuracy of $\approx 0.91$. Moreover, we intentionally 'sabotaged' the training by setting $\eta = 1 \times 10^{-5}$ such that SGD is unable to converge properly. This leads to networks with accuracies ranging from 0.38–0.65. A complexity measure should be capable of distinguishing both classes of networks. However, as Figure A.1 (top) shows, this is *not* the case for the clustering coefficient. Neural persistence (bottom), on the other hand, results in two regimes that can clearly be distinguished, with the trained networks having a significantly smaller variance.

### A.2  PROOF OF THEOREM 2

*Proof.* We may consider the filtration from Section 3.1 to be a subset selection problem with constraints, where we select $n$ out of $m$ weights. The neural persistence $\mathrm{NP}(G_k)$ of a layer thus only depends on the *selected* weights that appear as tuples of the form $(1, w_i)$ in $\mathcal{D}_k$. Letting $\widetilde{\mathbf{w}}$ denote the vector of selected weights arising from the persistence diagram calculation, we can rewrite neural persistence as $\mathrm{NP}(G_k) = \|\mathbb{1} - \widetilde{\mathbf{w}}\|_p$. Furthermore, $\widetilde{\mathbf{w}}$ satisfies $\|\mathbf{w}_{\min}\|_p \leq \|\widetilde{\mathbf{w}}\|_p \leq \|\mathbf{w}_{\max}\|_p$. Since all transformed weights are non-negative in our filtration, it follows that (note the reversal of the two terms)

$$\|\mathbb{1} - \mathbf{w}_{\max}\|_p \leq \mathrm{NP}(G_k) \leq \|\mathbb{1} - \mathbf{w}_{\min}\|_p \,, \tag{6}$$

and the claim follows. □

### A.3  ADDITIONAL VISUALIZATIONS AND ANALYSES FOR EARLY STOPPING

Due to space constraints and the large number of configurations that we investigated for our early stopping experiments, this section contains additional plots that follow the same schematic: the top row shows the differences in accuracy and epoch for our measure when compared to the commonly-used validation loss. Each cell in the heatmap corresponds to a single configuration of $b$ and $g$. In the heatmap of accuracy differences, blue represents parameter combinations for which we obtain a *higher* accuracy than validation loss for the same parameters; white indicates combinations for which we obtain the same accuracy, while red highlights combinations in which our accuracy decreases. Similarly, in the heatmap of epoch differences, green represents parameter combinations for which we stop *earlier* than validation loss for the same parameter. The scatterplots in

Section 4.2 show an 'unrolled' version of this heat map, making it possible to count how many parameter combinations result in early stops while also increasing accuracy, for example. The heatmaps, by contrast, make it possible to compare the behaviour of the two measures with respect to each parameter combination. Finally, the bottom row of every plot shows how many times each measure was triggered for every parameter combination. We consider a measure to be triggered if its stopping condition is satisfied prior to the last training epoch. Due to the way the parameter grid is set up, no configuration above the diagonal can stop, because $b + g$ would be larger than the total number of training epochs. This permits us to compare the 'slopes' of cells for each measure. Ideally, each measure should consist of a dark green triangle, as this would indicate that *parameter* configuration stops all the time.

**MNIST**   Please refer to Figures A.2 and A.3. The colours in the difference matrix of the top row are slightly skewed because in a certain configuration, our measure loses $0.8\%$ of accuracy when stopping. However, there are many other configurations in which virtually no accuracy is lost and in which we are able to stop more than four epochs earlier. The heatmaps in the bottom row again indicate that neural persistence is capable of stopping for more parameter combinations in general. We do not trigger as often for some of them, though.

**CIFAR-10**   Please refer to Figure A.4. In general, we observe that this data set is more sensitive with respect to the parameters for early stopping. While there are several configurations in which neural persistence stops with an increase of almost $10\%$ in accuracy, there are also scenarios in which we cannot stop training earlier, or have to train longer (up to $15$ epochs out of $80$ epochs in total). The second row of plots shows our measure triggers reliably for more configurations than validation loss. Overall, the scatterplot of all scenarios (Figure A.5) shows that most practical configurations are again located in $Q_2$ and $Q_3$. While we may thus find certain configurations in which we reliably outperform validation loss as an early stopping criterion, we also want to point out that our measures behaves correctly for many practical configurations. Points in $Q_1$, where we train *longer* and achieve a *higher* accuracy, are characterized by a high patience $g$ of approximately $40$ epochs and a low burn-in rate $b$, or vice versa. This is caused by the training for CIFAR-10, which does not reliably converge for FCNs. Figure A.6 demonstrates this by showing loss curves and the mean normalized neural persistence curves of five runs over training (loss curves have been averaged over all runs; standard deviations are shown in grey; we show the first half of the training to highlight the behaviour for practical early stopping conditions). For 'Fashion-MNIST', we observe that $\overline{\text{NP}}$ exhibits clear change points during the training process, which can be exploited for early stopping. For 'CIFAR-10', we observe a rather incremental growth for some runs (with no clearly-defined maximum), making it harder to derive a generic early stopping criterion that does not depend on fine-tuned parameters. Hence, we hypothesize that neural persistence cannot be used reliably in scenarios where the architecture is incapable of learning the data set. In the future, we plan to experiment with deliberately selected 'bad' and 'good' architectures in order to evaluate to what extent our topological measure is capable of assessing their suitability for training, but this is beyond the scope of this paper.

**IMDB**   Please refer to Figure A.7. For this data set, we observe that most parameter configurations result in *earlier* stopping (up to two epochs earlier than validation loss), with accuracy increases of up to $0.10\%$. This is also shown in the scatterplot A.8. Only a single configuration, viz. $g = 1$ and $b = 0$, results in a severe loss of accuracy; we removed it from the scatterplot for reasons of clarity, as its accuracy difference of $-21\%$ would skew the display of the remaining configurations too much (this is also why the legends do not include this outlier).

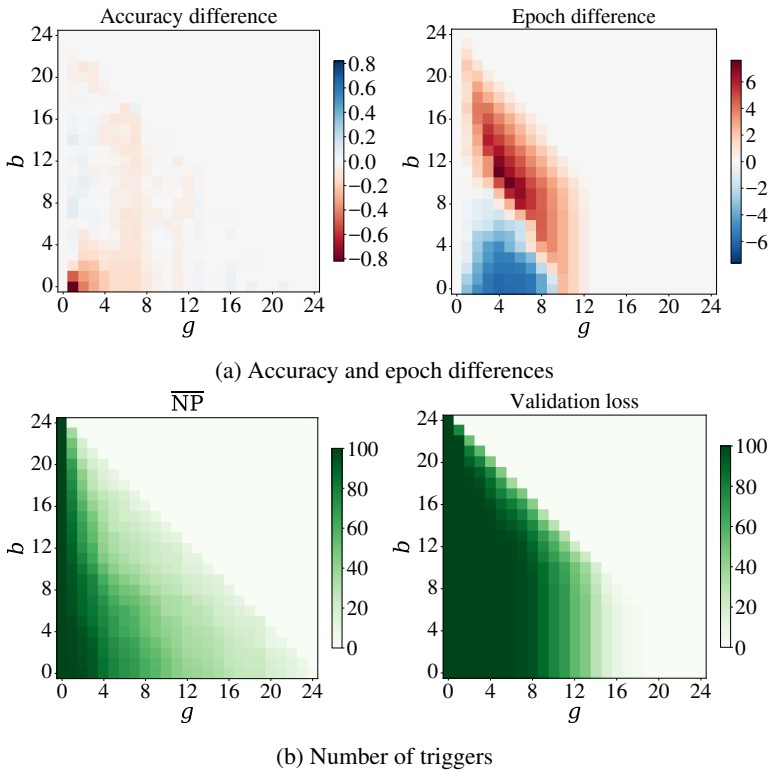

(a) Accuracy and epoch differences

(b) Number of triggers

Figure A.2: Additional visualizations for the 'MNIST' data set.

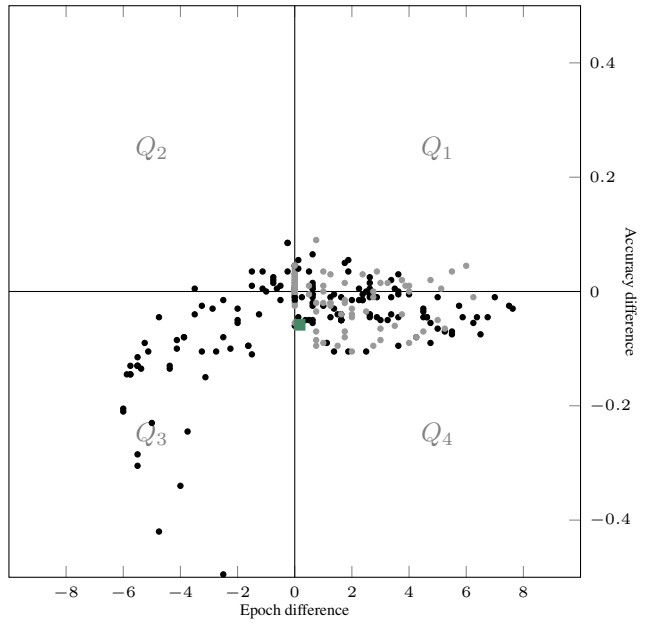

Figure A.3: Scatterplot of epoch and accuracy differences for 'MNIST'.

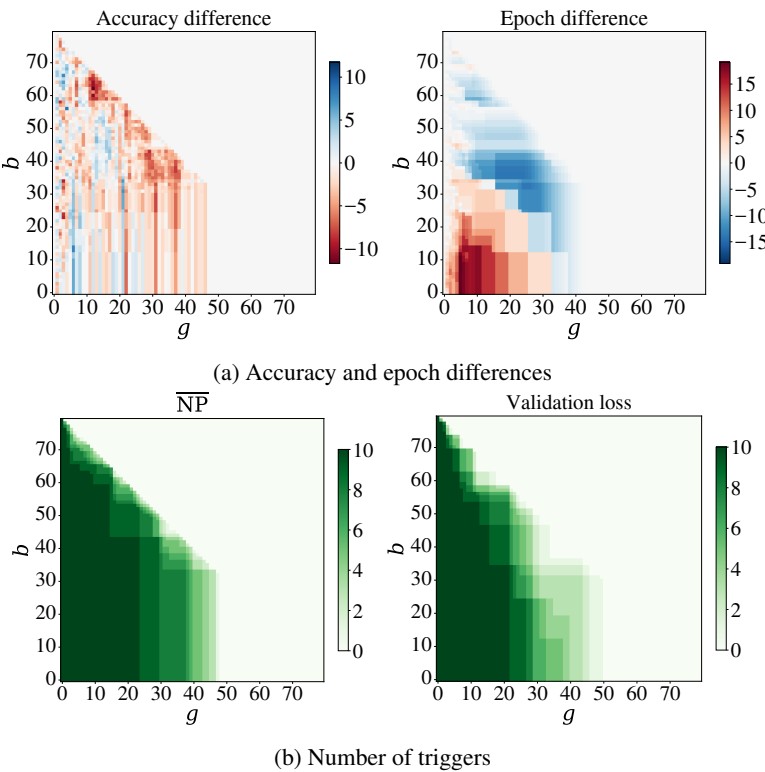

Figure A.4: Additional visualizations for the 'CIFAR-10' data set.

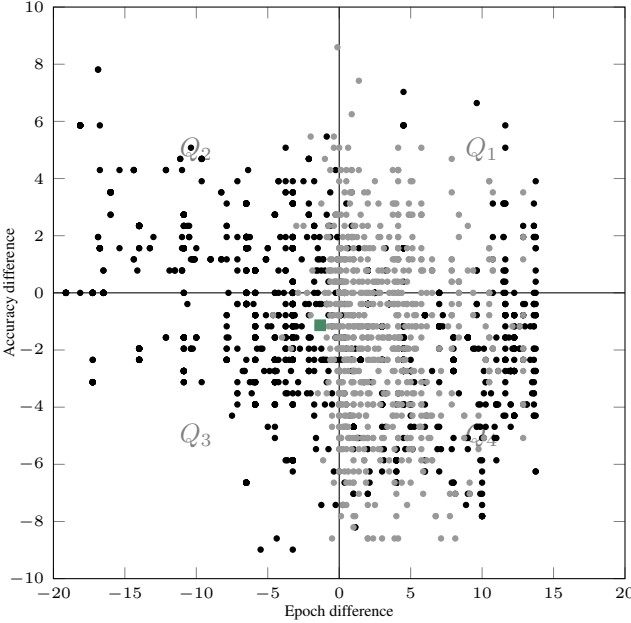

Figure A.5: Scatterplot of epoch and accuracy differences for 'CIFAR-10'.

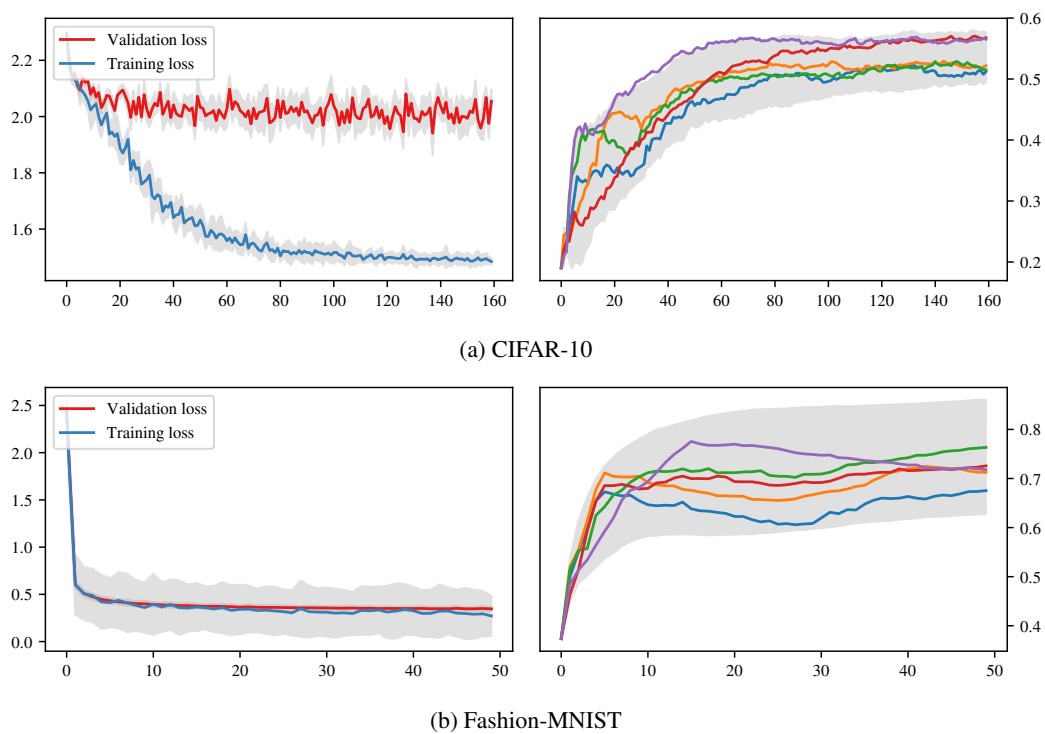

Figure A.6: A comparison of mean normalized neural persistence curves that we obtain during the training of 'CIFAR-10' and 'Fashion-MNIST'.

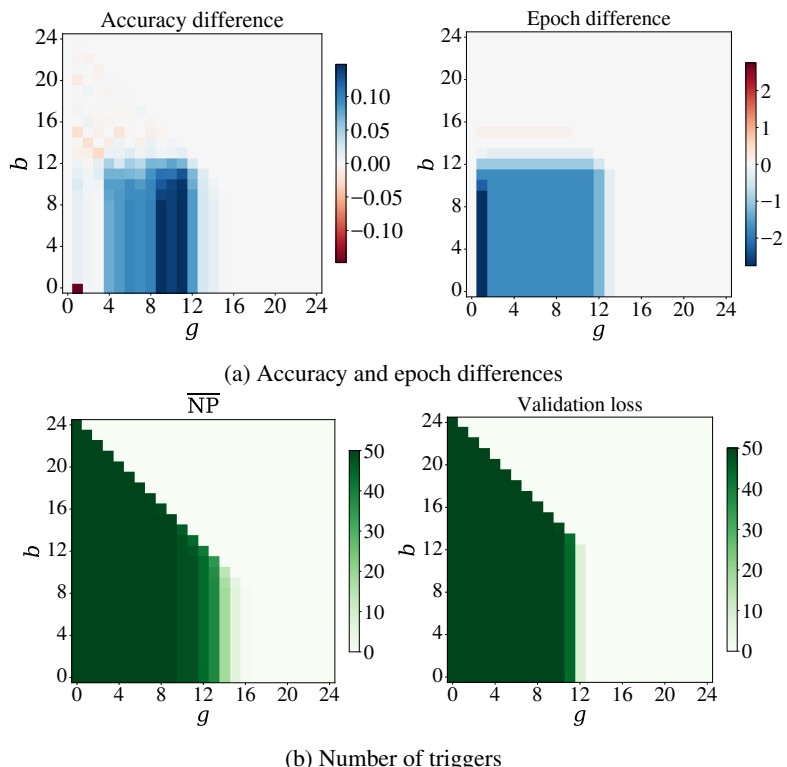

Figure A.7: Additional visualizations for the 'IMDB' data set.

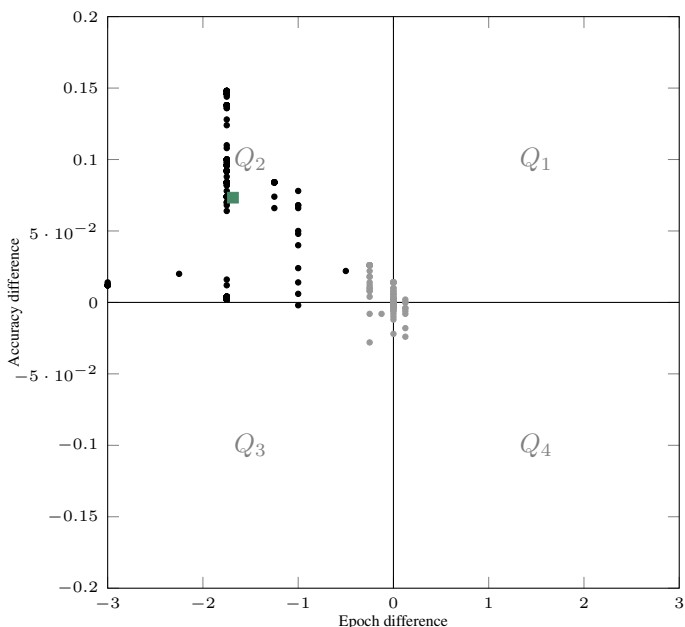

Figure A.8: Scatterplot of epoch and accuracy differences for 'IMDB'.

## A.4 NEURAL PERSISTENCE FOR CONVOLUTIONAL LAYERS

In principle, the proposed filtration process could be applied to any bipartite graph. Hence, we can directly apply our framework to convolutional layers, provided we represent them properly. Specifically, for layer $l$ we represent the convolution of its $i$th input feature map $a_i^{(l-1)} \in \mathbb{R}^{h_{\text{in}} \times w_{\text{in}}}$ with the $j$th filter $H_j \in \mathbb{R}^{p \times q}$ as one bipartite graph $G_{i,j}$ parametrized by a sparse weight matrix $W_{i,j}^{(l)} \in \mathbb{R}^{(h_{\text{out}} \cdot w_{\text{out}}) \times (h_{\text{in}} \cdot w_{\text{in}})}$, which in each row contains the $p \cdot q$ unrolled values of $H_j$ on the diagonal, with $h_{\text{in}} - p$ zeros padded in between after each $p$ values of $\text{vec}(H_j)$. This way, the flattened pre-activation can be described as $\text{vec}(z_{i,j}^{(l)}) = W_{i,j}^{(l)} \cdot \text{vec}(a_i^{(l-1)}) + b_{i,j}^l \cdot \mathbb{1}_{(h_{\text{out}} \cdot w_{\text{out}}) \times 1}$.

Since flattening does not change the topology of our bipartite graph, we compute the normalized neural persistence on this sparse weight matrix $W_{i,j}^{(l)}$ as the unrolled analogue of the fully-connected network's weight matrix. Averaging over all filters then gives a per-layer measure, similar to the way we derived mean normalized neural persistence in the main paper.

When studying the unrolled adjacency matrix $W_{i,j}^{(l)}$, it becomes clear that the edge filtration process can be approximated in a closed form. Specifically, for $m$ and $n$ input and output neurons we initialize $\tau = m + n$ connected components. When using zero padding, the additional dummy input neurons have to included in $m$. For all $\tau$ tuples in the persistence diagram the creation event $c = 1$. Notably, each output neuron shares the same set of edge weights.

Due to this, the destruction events—except for a few special cases—simplify to a list of length $\tau$ containing the largest filter values (each value is contained $n$ times) in descending order until the list is filled. This simplification of neural persistence of a convolution with one filter is shown as a closed expression in Equations 7–11, and our implementation is sketched in Algorithm 3. We thus obtain

$$\text{NP}(G_{i,j}) = \|\mathbb{1} - \widetilde{\mathbf{w}}\|_p, \tag{7}$$

where we use

$$\|\widetilde{\mathbf{w}}\|_p \leq \left\| \left(0, \mathbf{w}_c^T, \mathbf{w}_{\bar{c},\phi}^T, \text{vec}(A_\phi)^T, \text{vec}(B_\phi)^T \right)^T \right\|_p, \tag{8}$$

with

$$\phi = \tau - \dim(\mathbf{w}_c) - 1, \tag{9}$$
$$A_x = \mathbf{w}_{1:\lfloor \frac{x}{n} \rfloor} \otimes \mathbb{1}_{n-1}, \tag{10}$$
$$B_y = \mathbf{w}_{\lfloor \frac{y}{n} \rfloor + 1} \otimes \mathbb{1}_{y \bmod n}, \tag{11}$$

where $\mathbb{1}_0 := 0$. Following this notation, Equation 7 expresses neural persistence of the bipartite graph $G_{i,j}$, with $\widetilde{\mathbf{w}}$ denoting the vector of selected weights (i.e. the destruction events) when calculating the persistence diagram. We use $\mathbf{w}$ to denote the flattened and sorted weight values (in descending order) of the convolutional filter $H_j$, while $\mathbf{w}_c$ represents the vector of all weights that are located in a corner of $H_j$, whereas $\mathbf{w}_{\bar{c},\phi}$ is the vector of all weights which do *not* originate from the corner of the filter while still belonging to the first (and thus *largest*) $\lfloor \frac{\phi}{n} \rfloor$ weights in $\mathbf{w}$, which we denote by $\mathbf{w}_{1:\lfloor \frac{\phi}{n} \rfloor}$.

For the subsequent experiments (see below), we use a simple CNN that employs $32 + 2048$ filters. Hence, by using the shortcut described above, we do not have to unroll 2080 weight matrices explicitly, thereby gaining *both* in memory efficiency and run time, as compared to the naive approach: on average, a naive exact computation based on unrolling required $8.77$ s per convolutional filter and evaluation step, whereas the approximation only took about $0.000\,38$ s while showing very similar behaviour up to a constant offset.

For our experiments, we used an off-the-shelf 'LeNet-like' CNN model architecture (two convolutional layers each with max pooling and ReLU, 1 fully-connected and softmax) as described in Abadi et al. (2015). We trained the model on 'Fashion-MNIST' and included this setup in the early stopping experiments (100 runs of 20 epochs). In Figure A.9, we observe that stopping based on the neural persistence of a convolutional layer typically *only* incurs a considerable loss of accuracy: given a final test accuracy of $91.73 \pm 0.13$, stopping with this naive extension of our measure reduces accuracy by up to 4%. Furthermore, in contrast to early stopping on a fully-connected architecture,

---

**Algorithm 3** Approximating Neural Persistence of Convolutions per filter

---

**Require:** filter $H \in \mathbb{R}^{p \times q}$; number of input and output neurons as $m, n$

1: $\mathcal{T} \leftarrow \emptyset$           ▷ Initialize set of tuples for persistence diagram
2: $\tau \leftarrow m + n, \ t \leftarrow 0, \ i \leftarrow 0$    ▷ Initialize number of tuples, tuple counter, weight index
3: $h_{\max} \leftarrow \max_{h \in H} |h|$          ▷ Determine largest absolute weight
4: $H' \leftarrow \{|h|/h_{\max} \mid h \in H\}$        ▷ Transform weights for filtration
5: $s \leftarrow \text{sort}(\text{vec}(H'))$         ▷ Sort weights in descending order
6: $H'_c \leftarrow \{h'_{0,0}, h'_{0,q-1}, h'_{p-1,0}, h'_{p-1,q-1}\}$   ▷ Determine the set of all corner weights of filter $H'$
7: $\mathcal{T} \leftarrow (1,0), \ t \leftarrow t + 1$       ▷ Add tuple for surviving component
8: **for** $h'_c \in H'_c$ **do**         ▷ Each corner of $H'$ merges components
9:    $\mathcal{T} \leftarrow (1, h'_c), \ t \leftarrow t + 1$
10: **end for**
11: **while** 1 **do**        ▷ Create the remaining tuples (Approximation step)
12:    $n' = n - \text{Ind}(s[i] \in H'_c)$    ▷ if current weight is a corner weight, write one less tuple
13:    **if** $t + n' \leq \tau$ **then**    ▷ if there are at least $n'$ more tuples, set their merge value to $s[i]$
14:      **repeat** $n'$ times
15:        $\mathcal{T} \leftarrow (1, s[i])$   ▷ approximative as $s[i]$ does not always add $n'$ merges due to loops
16:      $t \leftarrow t + n', \ i \leftarrow i + 1$
17:    **else**          ▷ otherwise, process the remaining tuples similarly
18:      **repeat** $(\tau - t)$ times
19:        $\mathcal{T} \leftarrow (1, s[i])$
20:      **break**
21:    **end if**
22: **end while**
23: **return** $\|\mathcal{T}\|_p$       ▷ Compute norm of approximated persistence diagram

---

we do not observe any parameter combinations that stop early *and* increase accuracy. In fact, there is no configuration that results in an increased accuracy. This empirically confirms our theoretical scepticism towards naively applying our edge-focused filtration scheme to CNNs.

### A.5 RELATIONSHIP BETWEEN NEURAL PERSISTENCE AND VALIDATION ACCURACY

Motivated by Figure 2, which shows the different 'regimes' of neural persistence for a perceptron network, we investigate a possible correlation of (high) neural persistence with (high) predictive accuracy. For deeper networks, we find that neural persistence measures structural properties that arise from different parameters (such as training procedures or initializations), and *no* correlation can be observed.

For our experiments, we constructed neural networks with a *high* neural persistence prior to training. More precisely, following the theorems in this paper, we initialized most weights of each layer with very low values and reserved high values for very few weights. This was achieved by sampling the weights from a *beta distribution* with $\alpha = 0.005$ and $\beta = 0.5$. Using this procedure, we are able to initialize [20,20,20] networks with $\overline{\text{NP}} \approx 0.90 \pm 0.003$ compared to the same networks that have $\overline{\text{NP}} \approx 0.38 \pm 0.004$ when initialized by Xavier initialization. The mean validation accuracy of these untrained networks on the 'Fashion-MNIST' data set is $0.10 \pm 0.01$ and $0.09 \pm 0.03$, respectively.

Figure A.10 depicts how both types of networks converge to similar regimes of validation accuracy, while the mean normalized neural persistence achieved at the end of the training varies. For networks initialized with high $\overline{\text{NP}}$ (Figure A.10, left) the validation accuracy of networks with final $0.9 \leq \overline{\text{NP}} \leq 0.95$ ranges from $0.098$ (not shown) to $0.863$. For Xavier initialization (Figure A.10, right), the lack of correlation can also be observed. Furthermore, comparing the two plots, there are no clear advantages in initializing networks with high $\overline{\text{NP}}$. This observation further motivates the proposed *early stopping criterion*, which checks for *changes* in the $\overline{\text{NP}}$ value, and considers stagnating values to be indicative of a trained network.

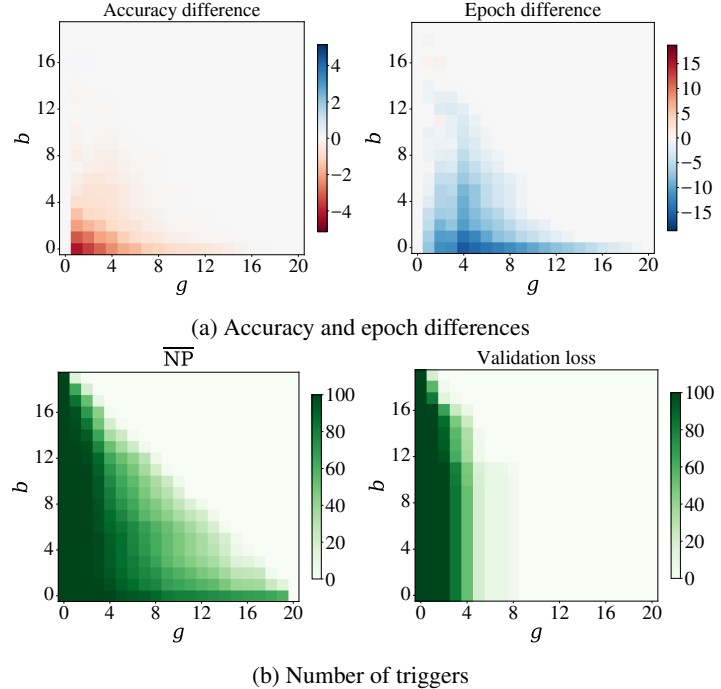

(a) Accuracy and epoch differences

(b) Number of triggers

Figure A.9: Additional visualizations for the 'Fashion-MNIST' data set, following the preliminary examination of convolutional layers. Here, the approximated neural persistence calculation for the first convolutional layer was used. However, we also ran few runs of the same experiment using the exact method which showed the same results. Employing the second convolutional layer or both did not improve this result.

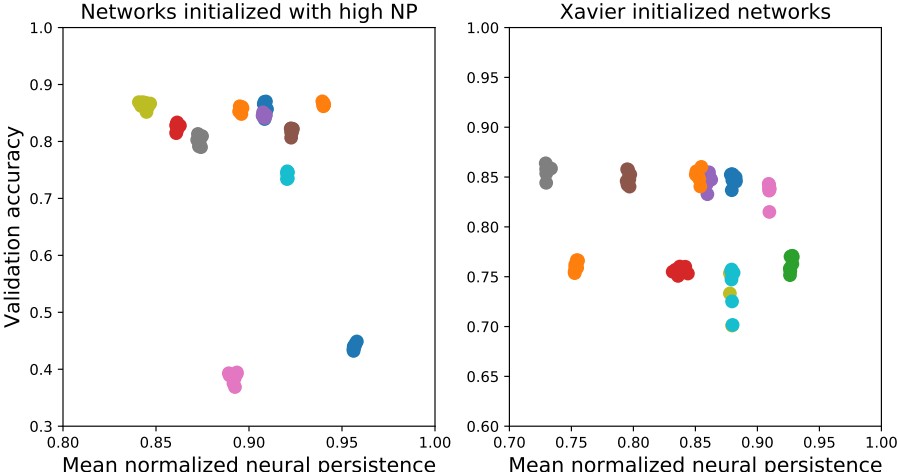

Figure A.10: Each cluster of points represent the last two training epochs (sampled every quarter epoch) of a [20,20,20] network trained on the 'Fashion-MNIST' data set. We observe no correlation between validation accuracy and normalized total persistence

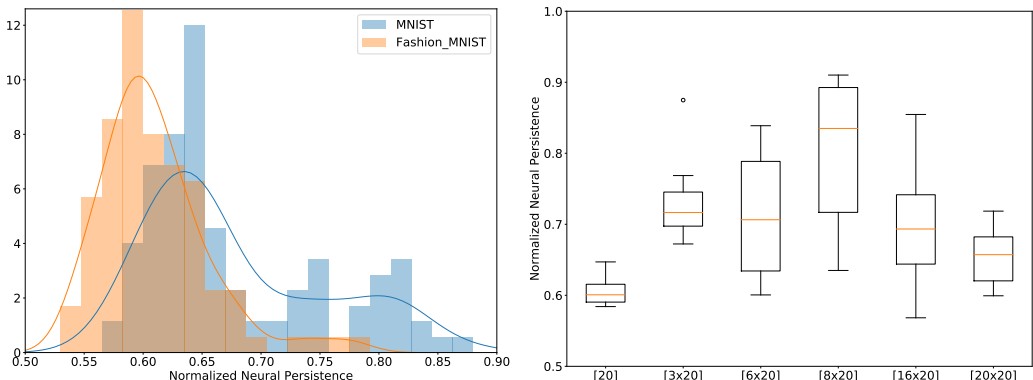

Figure A.11: (left) Histogram of the final normalized neural persistence of a $[50, 50, 20]$ network for 100 runs and 25 epochs of training. (right) Normalized neural persistence after 15 epochs of training on MNIST for different architectures with increasing depth. Deeper architectures are denoted as $[n \times 20]$ where $n$ is the number of hidden layers.

## A.6 NEURAL PERSISTENCE FOR DIFFERENT DATA DISTRIBUTIONS AND DEEPER FCN ARCHITECTURES

Neural persistence captures information about different data distributions during training. The weights tuned via backpropagation are directly influenced by the input data (as well as their labels) and neural persistence tracks those changes. To demonstrate this, we trained the same architecture, i.e. $[50, 50, 20]$, on two data sets with the same dimensions but different properties: MNIST and 'Fashion-MNIST'. Each data set has the same image size ($28 \times 28$ pixels, one channel) but lay on different manifolds. Figure A.11 (left) shows a histogram of the mean normalized neural persistence ($\overline{\text{NP}}$) after 25 epochs of training over 100 different runs. The distributions have a similar shape but are shifted, indicating that the two datasets lead the network to different topological regimes.

We also investigated the effect of depth on neural persistence. We selected a fixed layer size (20 hidden units) and increased the number of hidden layers. Figure A.11 (right) depicts the boxplots of mean $\overline{\text{NP}}$ for multiple architectures after 15 epochs of training on MNIST. Adding layers initially increases the variability of $\overline{\text{NP}}$ by enabling the network to converge to different regimes (essentially, there are many more valid configurations in which a trained neural network might end up in). However, this effect is reduced after a certain depth: networks with deeper architectures exhibit less variability in $\overline{\text{NP}}$.

## A.7 Early stopping in data scarcity scenarios

Labelled data is expensive in most domains of interest, which results in small data sets or low quality of the labels. We investigate the following experimental set-ups: (1) Reducing the training data set size and (2) Permuting a fraction of the training labels. We train a fully connected network ($[500, 500, 200]$ architecture) on 'MNIST' and 'Fashion-MNIST'. In the experiments, we compare the following measures for stopping the training: i) Stopping at the optimal test accuracy. ii) Fixed stopping after the burn in period. iii) Neural persistence patience criterion. iv) Training loss patience criterion. v) Validation loss patience criterion. For a description of the patience criterion, see Algorithm 2. All measures, except validation loss, include the validation datasets ($20\%$) in the training process to simulate a larger data set when no cross-validation is required. We report the accuracy on the non-reduced, non-permuted test sets. The batch size is 32 training instances. The stopping measures are evaluated every quarter epoch.

Figure A.12 shows the results averaged over 10 runs (the error is the standard deviation). The difference between the top and the bottom panel is the data set and the patience parameters. The $x$-axis depicts the fraction of the data set, which is warped for better accessibility. In each panel, the left-hand side subplots depict the results of the reduced data set experiment where the right-hand side subplots depict the result of the permutation experiments. The $y$-axis of the top subplot shows the accuracy on the non-reduced, non-permuted test set. The $y$-axis of the bottom subplot shows when the stopping criterion was triggered.

We note the following observations, which hold for both panels: More, non-permuted data yields higher test accuracy. Also, as expected, the optimal stopping gives the highest test accuracy. The fixed early stopping results in inferior test accuracy when only a fraction of the data is available. The neural persistence based stopping is triggered late when only a fraction of the data is available which results in a slightly better test accuracy compared to training and validation loss. The training loss stopping achieves similar test accuracies compared to the persistence based stopping (for all regimes except the very small data set) with shorter training, on average. We note that, it is generally not advisable to use training loss as a measure for stopping because the stability of this criterion also depends on the batch size. When only a fraction of the data is available, the validation loss based stopping stops on average after the same number of training epochs as the training loss, which results in inferior test accuracy because the network has seen in total fewer training samples. Most strikingly, validation loss based stopping is is triggered later (sometimes never) when most training and validation labels are randomly permuted which results in overfitting and poor test accuracy.

To conclude, the neural persistence based stopping achieves good performance without being affected by the batch size and noisy labels. The authors also note that the result is consistent for multiple architectures and most patience parameters.

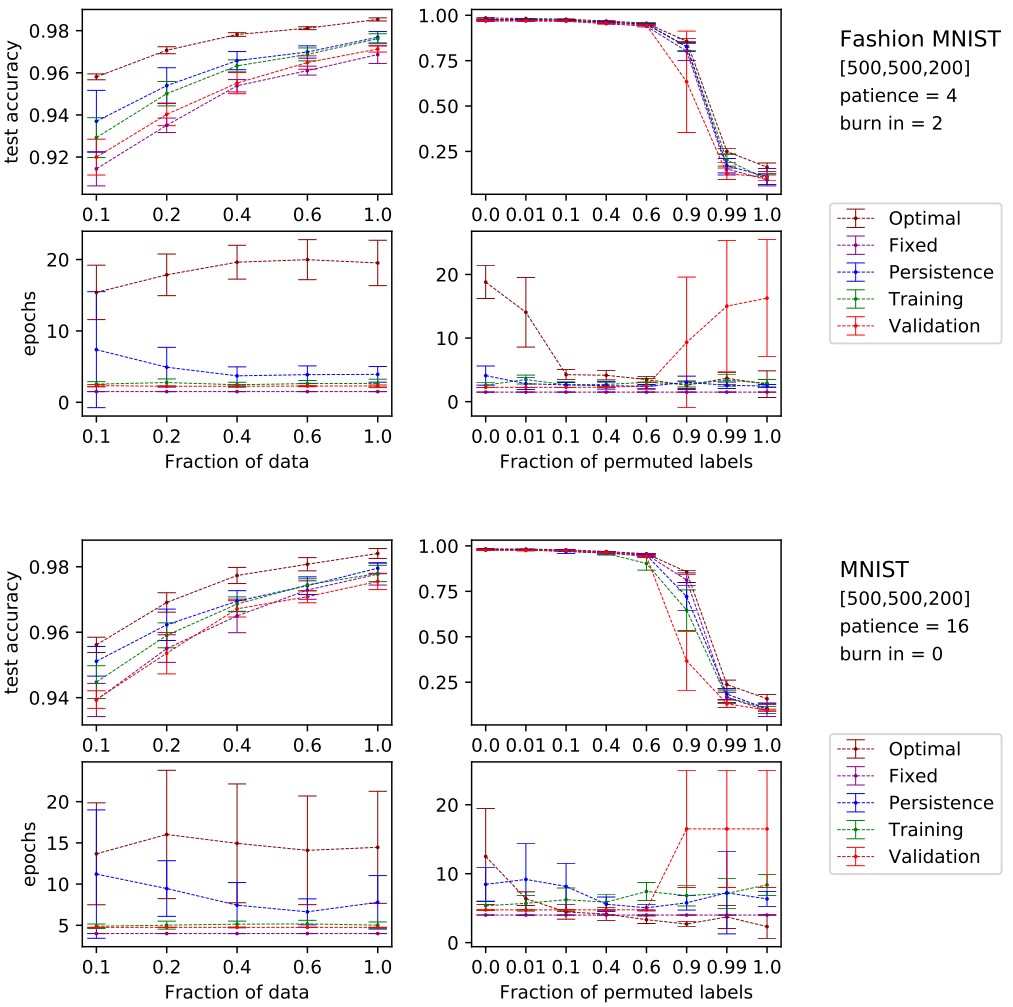

Figure A.12: On MNIST and Fashion-MNIST $\overline{\text{NP}}$ (in blue) stops later than validation and training loss when fewer training samples are available (left-hand side) which results in a higher test accuracy. For increasing noise in the training labels (right-hand side), the stopping of $\overline{\text{NP}}$ remains stable, in contrast to the validation loss stopping, which leads to lower test accuracy after longer training at a high fraction of permuted labels. The patience and burn in parameters are reported in quarter epochs.

Table A.1: Parameters and hyperparameters for the experiment on best practices and neural persistence. Dropout and batch normalization were applied after the first hidden layer. Throughout the networks, *ReLU* was the activation function of choice.

| Data set | # Runs | # Epochs | Architecture | Optimizer | Batch Size | Hyperparameters |
|---|---|---|---|---|---|---|
| MNIST | 50 | 40 | [650, 650] | Adam | 32 | $\eta = 0.0003\ \beta_1 = 0.9,\ \beta_2 = 0.999,\ \epsilon = 1 \times 10^{-8}$ 
 $\eta = 0.0003\ \beta_1 = 0.9,\ \beta_2 = 0.999,\ \epsilon = 1 \times 10^{-8}$, Batch Normalization 
 $\eta = 0.0003\ \beta_1 = 0.9,\ \beta_2 = 0.999,\ \epsilon = 1 \times 10^{-8}$, Dropout 50% |

Table A.2: Parameters and hyperparameters for the experiment on early stopping. Throughout the networks, *ReLU* was the activation function of choice.

| Data set | # Runs | # Epochs | Architecture | Optimizer | Batch Size | Hyperparameters |
|---|---|---|---|---|---|---|
| (Fashion-)MNIST | 100 | 10 | Perceptron | Minibatch SGD | 100 | $\eta = 0.5$ |
| | | 40 | [50, 50, 20] 
 [300, 100] 
 [20, 20, 20] | Adam | 32 | $\eta = 0.0003\ \beta_1 = 0.9,\ \beta_2 = 0.999,\ \epsilon = 1 \times 10^{-8}$ |
| CIFAR-10 | 10 | 80 | [800, 300, 800] | Adam | 128 | $\eta = 0.0003\ \beta_1 = 0.9,\ \beta_2 = 0.999,\ \epsilon = 1 \times 10^{-8}$ |
| IMDB | 5 | 25 | [128, 64, 16] | Adam | 128 | $\eta = 1 \times 10^{-5}\ \beta_1 = 0.9,\ \beta_2 = 0.999,\ \epsilon = 1 \times 10^{-8}$ |

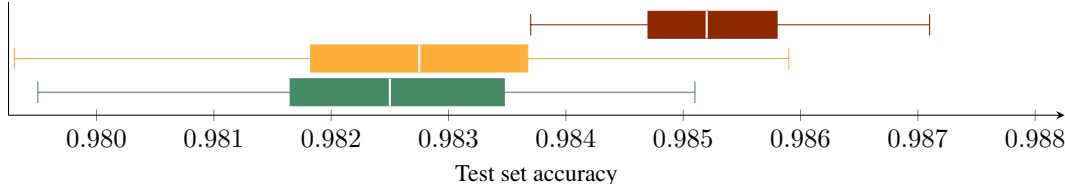

Figure A.13: Comparison of test set accuracy for trained networks without modifications (green), with batch normalization (yellow), and with 50% of the neurons dropped out during training (red) for the MNIST data set.

## A.8 TESTING ACCURACY OF DIFFERENTLY REGULARIZED MODELS

We showed in the main text that neural persistence is capable of distinguishing between networks trained with/without batch normalization and/or dropout. Figure A.13 additionally shows test set accuracies.

