# OpenReview forum: "Neural Persistence: A Complexity Measure for Deep Neural Networks Using Algebraic Topology"
_ICLR.cc/2019/Conference_

### Official Review · AnonReviewer1 · 2018-11-02
**An interesting new measure of network complexity**

**Rating:** 7
**Confidence:** 4

**Review:**

The authors, motivated by work in topological graph analysis, introduce a new broadly applicable complexity measure they call  neural persistence--essentially a sum over norms of persistence diagrams (objects from the study of persistent homology).  The also provide experiments testing their parameter, primarily on MNIST with some work on CIFAR-10.

I'd like to preface my criticism with the following: this work is extremely compelling, and the results and experiments are sound.  I'm very interested to see where this goes.  Figure 2 is particularly compelling!

That said, I am extremely suspicious of proposals for measures of generalization which (1) do not make contact with the data distribution being studied, and (2) which are only tested on MNIST and CIFAR-10.  Additionally, (3) it is not clear what a "good" neural persistence is, a priori, and (4) I'm not entirely sure I agree with the author's assessment of their numerical data.

In more detail below:

1. At this point, there's a tremendous number of different suggested ways to measure "generalization" by applying different norms and bounds and measures from all of the far reaches of mathematics.  A new proposed measure **really needs** to demonstrate a clear competitive measure against other candidates.  The authors make a strong case that this measure is better than competitors from TGA, but I'm not yet convinced this measure is doing enough legwork.  For example, is it possible that a network has high neural persistence, but still has terrible test or validation error?  Why or why not?  Are there obvious counterexamples?  Are there reasons to think those obvious counterexamples aren't like trained neural networks?  These are all crucial questions to ask and answer if you want this sort of measure to be taken seriously.

2.  Most of your numerical experiments were on MNIST, and MNIST is weird.  It's getting to be a joke now in the community that your idea works on MNIST, but breaks once you try to push it to something harder.  Even Cifar-10 has its quirks, and observations that are true of some networks absolutely do not generalize to others.

3. While I'm convinced that neural persistence allows you to distinguish between networks trained in different ways, it isn't clear why I should expect a particular neural persistence to mean anything at all w.r.t. validation loss.  Are there situations in which the neural persistence has stopped changing, but validation loss is still changing appreciably?  Why or why not?

4. I'm concerned that the early stopping procedure used as a benchmark wasn't tuned as carefully as neural persistence was.  I also honestly cannot determine anything from Figure 4 except that your "Fixed" baseline is bad, and that persistence seems to do about the same as validation loss.  It even seems that Training loss is a better early stopping criteria (better than both validation and persistence!) from this plot, because it seems to perform just as well, and systematically stop earlier.  Am I reading this plot right (particularly for 1.0 fraction MNIST)?


This work currently seems like a strong candidate for the workshop track.  I would have difficulty raising my score above much more than a 6 without much more numerical data, and analysis of when the measure fails.

Edit: The authors have made a significant effort to address my concerns, and I'm updating my score to 7 from 5 in response.

---

> ### Author Response · Authors · 2018-11-25
> **Our response to your review**
>
> We would like to thank the reviewers for their valuable insights and remarks that we address individually below. We significantly extended the paper and the supplementary materials, focusing particularly (as suggested by reviewers 1 and 3) on providing a thorough analysis and evaluation of our early stopping criterion. Moreover, as recommended by R1, we now discuss additional data sets. Due to the requested changes, we updated Section 4.2 and show the ‘Fashion-MNIST’ data set (in the main paper), while describing results for other data sets in the appendix.
>
> To summarize the changes:
>
> - In Section 4.2, we conducted a detailed analysis of our early stopping criterion for different sets of hyperparameters and different data sets (three image classification data sets: MNIST, CIFAR-10, Fashion-MNIST, and one text classification dataset: IMDB). Our stopping criterion generalizes well to other situations and is competitive with validation loss-based stopping criteria.
>
> - We extended the theoretical section (Section A.4) to include details of normalized neural persistence computation on convolutional layers. Moreover, we describe preliminary experiments about early stopping based on convolutional layers.
>
> - We describe properties and limitations of the measure (Section A.5): initialization of networks with high neural persistence does not, as expected, correlate with higher accuracy, for example.
>
> - We describe the behaviour of neural persistence for deep architectures as well as its
> relationship with the learned data distribution (Section A.6 in the appendix).
>
> - We extended the discussion of early stopping in scenarios with scarce data to show that our measure behaves as expected, i.e. it stops earlier when overfitting can occur, and it stops later when longer training is beneficial (Section A.7).
>
> Individual answers to your review:
>
> We’d like to address the four points you raised as follows:
>
> > At this point, there's a tremendous number of different suggested ways to measure "generalization" [...].  A new proposed measure **really needs** to demonstrate a clear competitive measure against other candidates.  The authors make a strong case that this measure is better than competitors from TGA, but I'm not yet convinced this measure is doing enough legwork.  For example, is it possible that a network has high neural persistence, but still has terrible test or validation error?  Why or why not?  Are there obvious counterexamples? [...]
>
> First, Section A.6 shows that neural persistence makes contact with the data distribution. We also extended the evaluation: Section A.5 analyses situations where high neural persistence and terrible validation error co-occur. In brief, these situations can only occur after artificially initializing the network with high neural persistence. Section A.7 shows the legwork: Compared to validation loss, neural persistence stops earlier when overfitting can occur and it stops later when longer training is beneficial.
>
> > Most of your numerical experiments were on MNIST, and MNIST is weird. [...]
>
> We acknowledge your concerns, so we added another image (Fashion-MNIST) and a text classification data set (IMDB Large Movie Review Dataset). Figure 4 and Section A.3 show a quantitative summary over all data sets and networks, including CNNs. We observed that our early stopping criterion is competitive in all scenarios.
>
> > While I'm convinced that neural persistence allows you to distinguish between networks trained in different ways, it isn't clear why I should expect a particular neural persistence to mean anything at all w.r.t. validation loss.  Are there situations in which the neural persistence has stopped changing, but validation loss is still changing appreciably?  Why or why not?
>
> A “good” neural persistence is not defined a priori, and this is why we couple the stabilization of our measure with the training procedure. Figure A.12, rhs shows that for noisy labels, neural persistence stabilizes soon where validation loss is still decreasing.
>
> > I'm concerned that the early stopping procedure used as a benchmark wasn't tuned as carefully as neural persistence was.  I also honestly cannot determine anything from Figure 4 except that your "Fixed" baseline is bad, and that persistence seems to do about the same as validation loss. [...] Am I reading this plot right (particularly for 1.0 fraction MNIST)?
>
> For a qualitative evaluation of all parameter choices of the stopping criteria please see Figure 4 where neural persistence shows competitive performance for all scenarios. Section A.7 extends the findings on freeing validation data to a second data set and to a second real-world scenario. This includes the following point: Training loss stops earlier than neural persistence with only slightly lower performance on the test set. However, training loss fluctuates with varying the batch size and has no theoretical guarantees. (you were reading the plot right)

---

> > ### Comment · AnonReviewer1 · 2018-11-29
> > **Updated score**
> >
> > I thank the authors for their extensive work to address my concerns.  I've updated my score to a 7 in response.
> >
> > One small concern is that I'm not sure the claims regarding Fig. 11 are entirely justifiable given the fairly large error bars.  This is a comparatively minor point, but softening the language / more empirical data to reduce the error bars would be great.

---

> > > ### Author Response · Authors · 2018-12-03
> > > **Thank you!**
> > >
> > > Thank you for this very positive change! We aim to update the discussion concerning Fig. 11 in a revision of the paper.

---

### Official Review · AnonReviewer2 · 2018-11-05
**An interesting idea, but insufficient.**

**Rating:** 4
**Confidence:** 4

**Review:**

This paper proposes to analyze the complexity of a neural network using its zero-th persistent homology. Each layer is considered a bipartite graph with edge weights. As edges are being added in a monotonically decreasing order, each time a connected component is merged with others will be recorded as a new topological feature. The persistence of each topological feature is measured as the weight difference between the new edge and the maximal weight (properly normalized). Experiments show that by monitoring the p-norm of these persistence values one can stop the training a few epochs earlier than the validation-error-based early stopping strategy, with only slightly worse test accuracy.

The proposed idea is interesting and novel. However, it is needs a lot of improvement for the following reasons.

1) The proposed idea can be explored much deeper. Taking a closer look, these zero-th persistence are really the weights of the maximum spanning tree (with some linear transformation). So the proposed complexity measure is really the p-norm of the MST. This raises other related questions: what if you just take all the weights of all edges? What if you take the optimal matching of the bipartite graph? How about the top K edges? I am worried that the p-norms of these edge sets might have the same effect; they converge as the training converges. These different measurements should be at least experimentally compared in order to show that the proposed idea is crucial.

Note also that most theoretical proofs are straightforward based on the MST observation.

2) The experiment is not quite convincing. For example, what if we stop the training as soon as the improvement of validation accuracy slows down (converges with a much looser threshold)? Wouldn’t this have the same effect (stop slightly earlier with only slightly worse testing accuracy)? Also shouldn’t the aforementioned various alternative norms be compared with?

3) Some other ideas/experiments might worth exploring: taking the persistence over the whole network rather than layer-by-layer, what happens with networks with batch-normalization or dropout?

---

> ### Author Response · Authors · 2018-11-25
> **Our response to your review**
>
> We would like to thank the reviewers for their valuable insights and remarks that we address individually below. We significantly extended the paper and the supplementary materials, focusing particularly (as suggested by reviewers 1 and 3) on providing a thorough analysis and evaluation of our early stopping criterion. Moreover, as recommended by R1, we now discuss additional data sets. Due to the requested changes, we updated Section 4.2 and show the ‘Fashion-MNIST’ data set (in the main paper), while describing results for other data sets in the appendix.
>
> To summarize the changes:
>
> - In Section 4.2, we conducted a detailed analysis of our early stopping criterion for different sets of hyperparameters and different data sets (three image classification data sets: MNIST, CIFAR-10, Fashion-MNIST, and one text classification dataset: IMDB). Our stopping criterion generalizes well to other situations and is competitive with validation loss-based stopping criteria.
>
> - We extended the theoretical section (in the supplementary materials, Section A.4) to include details of normalized neural persistence computation on convolutional layers. Moreover, we describe preliminary experiments about early stopping based on convolutional layers.
>
> - We describe properties and limitations of the measure (in the supplementary materials, Section A.5): initialization of networks with high neural persistence does not, as expected, correlate with higher accuracy, for example.
>
> - We describe the behaviour of neural persistence for deep architectures as well as its
> relationship with the learned data distribution (Section A.6 in the appendix).
>
> We extended the discussion of early stopping in scenarios with scarce data to show that our measure behaves as expected, i.e. it stops earlier when overfitting can occur, and it stops later when longer training is beneficial (Section A.7).
>
> Individual answers to your review:
>
> Thanks for the comments and suggestions! In light of your review, we have restructured the paper to highlight certain details more prominently.
>
> > [...] This raises other related questions: what if you just take all the weights of all edges?  [...] I am worried that the p-norms of these edge sets might have the same effect; they converge as the training converges.
>
> We clarify this relationship in Section 3.1. Our measure is linked to the $p$-norm of the top n edges (whose linear transformation is in fact used for our lower bound in Theorem 2) and is therefore also somewhat related to the $p$-norm of all the weights. We originally discussed the $p$-norm of all weights in Section 4.2; we have since moved this item to the discussion to make it more prominent.
>
> The $p$-norm of all weights is not a valid early stopping criterion as it was never triggered earlier in our experiments (it only works for perceptrons); our neural persistence measure is thus an earlier marker of training convergence.
>
> > The experiment is not quite convincing. For example, what if we stop the training as soon as the improvement of validation accuracy slows down (converges with a much looser threshold)? [...]
>
> As a response to the reviews, we performed a broader evaluation of the early stopping criterion. Concerning the looser threshold, we discuss in the paper (Section 4.2) that we refrained from tuning this parameter as it is highly sensitive to the scale of the monitored measure (making it harder to compare fairly).  Currently, our evaluation follows the parameters of the Keras callback, which sets `min_delta` to zero by default.
>
> > Some other ideas/experiments might worth exploring: taking the persistence over the whole network rather than layer-by-layer, what happens with networks with batch-normalization or dropout?
>
> We initially considered integrating persistence of the whole network but left it for future work: preliminary results showed some promise but also indicated the need for a more complicated setup: we need to account for higher-order topological features and ensure that features are not being "masked" due to different scales. When using information of the whole network in a straightforward extension of our method, we observe that this is sufficient for describing shallow networks only. A new experiment of neural persistence for deeper architectures (Section A.5) shows that variability is not a simple function of depth, making it clear that the contributions of each layer need to be carefully considered if we want to have one filtration over the whole network. It is possible that Dowker complexes (https://arxiv.org/pdf/1608.05432.pdf) may be useful here because they are capable of capturing directions, but due to time constraints we were unable to pursue this idea during the rebuttal phase.
>
> As for the batch normalization and the dropout, we observed in Fig. 3 of the paper that while there is almost no difference between regular training and batch normalization, dropout generally increases the measured neural persistence.

---

### Official Review · AnonReviewer3 · 2018-11-06

**Rating:** 6
**Confidence:** 5

**Review:**

The paper proposes the notion of "neural persistence", i.e., a topological measure to assign scores to fully-connected layers in a neural network. Essentially, a simplicial complex is constructed by considering neurons as 0-simplices and connections as 1-simplices. Using the (normalized) connection weights then facilitates to define a filtration. Persistent homology (for 0-dim. homology groups) then provides a concise summary of the evolution of the 0-dim. features over the filtration in the form of a barcode. The p-norm of the persistence diagram (containing points (1,w_i)) is then used to define the "neural persistence"  NP(G_k) of a layer G_k; this measure is averaged over all layers to obtain one final neural persistence score. Thm. 1 establishes lower and upper bounds on N(G_k); Experiments show that neural persistence, measured for small networks on MNIST, aligns well with previous observations that batch-norm and dropout are benefical for generalization and training. Further, neural persistence it can be used as an early stopping criterion without having to rely on validation data.

Overall, I think this is an interesting and well-written paper with a good overview of related work in terms of using TDA approaches in machine learning. The theoretical aspects of the work (i.e., the bounds) are fairly obvious. The bounds
are required, though, for proper normalization. Using 0-dim. persistent homology is also appropriate in this context, as I tend to agree with the authors that this aspect is the most interesting one (and also the only computationally feasible one if this needs to be done during training).

The only major concern at this point, is the experimental evaluation on small fully-connected networks.
While reading the paper, I was wondering how this could be generalized, e.g., to convolution layers, as the strategy seems to be also applicable in this context as well. I do think that the results on MNIST are convincing, however, already on CIFAR-10 the early stopping criterion seems to be very sensitive to the choice of g (from what I understood). So, this raises the obvious question of how this behaves for larger networks with more layers and larger datasets. If the contribution boils down to a confirmation that dropout and batch-norm are beneficial, this would substantially weaken the paper. Specifically, I would be interested in having full-connected networks with more layers (possibly less neurons per layer). Maybe the authors can comment on that or perform experiments along this direction.

Minor comments:

- What is the subscript d in \mathcal{D}_d intended to denote?
- In Thm.1 - why should \phi_k be unique? This is not the only choice?
- End of Sec. 4. - "it is beneficial to free validation data ..." - What does that mean?

---

> ### Author Response · Authors · 2018-11-25
> **Our response to your review**
>
> We would like to thank the reviewers for their valuable insights and remarks that we address individually below. We significantly extended the paper and the supplementary materials, focusing particularly (as suggested by R1/R3) on providing a thorough analysis/evaluation of our early stopping criterion. As recommended by R1, we discuss additional data sets. Given the requested changes, we updated Section 4.2 and show the ‘Fashion-MNIST’ data set in the paper, while describing other results in the appendix.
>
> Summary of changes:
>
> - In Section 4.2, we added a detailed analysis of our early stopping criterion for different parameters and data sets (MNIST, CIFAR-10, Fashion-MNIST, IMDB). Our criterion generalizes well and is competitive with validation loss-based stopping criteria.
>
> - We added a theoretical section (Section A.4) on conv layers, and describe preliminary experiments about early stopping for CNNs.
>
> - We describe properties and limitations (Section A.5): initialization of networks with high neural persistence does not, as expected, correlate with higher accuracy, for example.
>
> - We describe the behaviour of neural persistence for deep architectures plus its relationship with the data distribution (Section A.6).
>
> - We extended the discussion of early stopping in data scarce scenarios: we stop earlier when overfitting can occur, and we stop later when longer training is beneficial (Section A.7).
>
> Individual answers to your review:
>
> Thanks for your comments and questions! We prepared a new set of experiments to address them.
>
> > [...] how this could be generalized, e.g., to convolution layers [...]
>
> We updated the appendix with a Section on conv layers (A.4). Two observations arise: (1) there's a computational issue; 'unrolling' each filter into a weight matrix requires more time than for FCNs; we sketch a new approximative algorithm. (2) our filtration focuses on the edge neighbourhood of vertices (neurons), which is relatively redundant in CNNs, so our current method does not capture the relevant topology of a CNN. We hypothesize that we should include activations and plan to investigate this in future work.
>
> Despite these hurdles, we added a new experiment on CNNs: we unrolled each conv filter into a graph, computing our measure per filter, and summed over all filters of a layer (corresponding to our setting in FNCs). We exploit redundancy of filter values, which leads to a simplification of our filtration and an approximation of NP (see Algorithm 3).
>
> In our early stopping experiments we see that our measure performs better at early stopping on FCNNs as compared to CNNs, which empirically confirms our theoretical scepticism towards directly applying our edge-focused filtration to CNNs.
>
> > I do think that the results on MNIST are convincing, however, already on CIFAR-10 [...] seems to be very sensitive to the choice of g [...]. So, this raises the obvious question of how this behaves for larger networks with more layers and larger datasets.
>
> We have updated our previous setup to be more inclusive. We perform an extensive evaluation for early stopping on different data sets. For CIFAR-10, we can show that both measures (val. loss and neural pers.) are sensitive to the parameters. Figure A.5 shows that there a good scenarios for each of the measures. Out of all the data sets, our epoch/accuracy differences are worst here; we are comparable, if not better than, validation loss in terms of parameter sensitivity, though. Plus, the new experiment permits us to link performance back to the training itself (Figure A.6). There appears to be a relation between mediocre training performance of an FCN on CIFAR-10 and the mediocre early stopping behaviour. In the future, we'd like to properly extend our method to CNNs; this has challenges that need to be addressed outside the scope of this paper.
>
> > Specifically, I would be interested in having full-connected networks with more layers [...]
>
> We have performed additional experiments for deeper neural networks: we observe an interesting relationship between network depth and the variability of our measure (Section A.6).
>
> > What is the subscript $d$ in $\mathcal{D}_d$ intended to denote?
>
> This indicates the dimension of the corresponding persistence diagram.
>
> > In Thm.1 - why should \phi_k be unique? [...]
>
> We require only one function that returns the current weight of an edge. We have since removed the adjective for clarity.
>
> > End of Sec. 4. - "it is beneficial to free validation data ..." - What does that mean?
>
> In deep learning applications, where sample size is critical, it can make a difference to free X% of the samples normally used for validation and use them for training instead. Accuracy and generalization capabilities were shown to be highly dependent on the amount of training data. We have since rewritten that section (end of Section 4.2) and included a new set of experiments (Section A.7 in the appendix).

---

### Public Comment · (anonymous) · 2018-10-07
**Missing citation**

To whom it may concern,

It appears that this paper is missing a relevant citation for recent work in measures of capacity for neural networks using algebraic topology: "On Characterizing the Capacity of Neural Networks using Algebraic Topology" ( https://arxiv.org/abs/1802.04443 ).

---

> ### Author Response · Authors · 2018-10-08
> **Is there a more recent version available?**
>
> Thanks for pointing this out! At the time of us writing this paper, the article that you mentioned was not published in a peer-reviewed venue. Is there a more recent version that we missed?

---

> ### Author Response · Authors · 2018-11-25
> **Fixed in the revised version**
>
> Thanks for the input; in the revised version, we have included this reference.

---

### Meta-Review · Area_Chair1 · 2018-12-15
**A topological complexity measure of neural networks based on persistent 0-homology of weights, with a new early stopping criterion.**

**Confidence:** 5
**Recommendation:** Accept (Poster)

**Metareview:**

The paper presents a topological complexity measure of neural networks based on persistence 0-homology of the weights in each layer. Some lower and upper bounds of the p-norm persistence diagram are derived that leads to normalized persistence metric. The main discovery of such a topological complexity measure is that it leads to a stability-based early stopping criterion without a statistical cross-validation, as well as distinct characterizations on random initialization, batch normalization and drop out. Experiments are conducted with simple networks and MNIST, Fashion-MNIST, CIFAR10, IMDB datasets.

The main concerns from the reviewers are that experimental studies are still preliminary and the understanding on the observed interesting phenomenon is premature. The authors make comprehensive responses to the raised questions with new experiments and some reviewers raise the rating.

The reviewers all agree that the paper presents a novel study on neural network from an algebraic topology perspective with interesting results that has not been seen before. The paper is thus suggested to be borderline lean accept.